# Nuclear Receptors and the Hidden Language of the Metabolome

**DOI:** 10.3390/cells13151284

**Published:** 2024-07-31

**Authors:** Yujie Chen, Matthew Tom Anderson, Nathaniel Payne, Fabio R. Santori, Natalia B. Ivanova

**Affiliations:** 1Center for Molecular Medicine, University of Georgia, Athens, GA 30602, USA; yc24786@uga.edu (Y.C.); matthew.anderson7699@gmail.com (M.T.A.); nathanieljusticepayne@gmail.com (N.P.); 2Department of Biochemistry and Molecular Biology, University of Georgia, Athens, GA 30602, USA

**Keywords:** nuclear receptor, hormones, metabolites, evolution, regulation, ligands

## Abstract

Nuclear hormone receptors (NHRs) are a family of ligand-regulated transcription factors that control key aspects of development and physiology. The regulation of NHRs by ligands derived from metabolism or diet makes them excellent pharmacological targets, and the mechanistic understanding of how NHRs interact with their ligands to regulate downstream gene networks, along with the identification of ligands for orphan NHRs, could enable innovative approaches for cellular engineering, disease modeling and regenerative medicine. We review recent discoveries in the identification of physiologic ligands for NHRs. We propose new models of ligand-receptor co-evolution, the emergence of hormonal function and models of regulation of NHR specificity and activity via one-ligand and two-ligand models as well as feedback loops. Lastly, we discuss limitations on the processes for the identification of physiologic NHR ligands and emerging new methodologies that could be used to identify the natural ligands for the remaining 17 orphan NHRs in the human genome.

## 1. Introduction

Nuclear hormone receptors (NHRs) play a central role in the development of the central nervous system, heart, lymphatics and blood vessels as well as in the regulation of the endocrine system and immune function. Their role in physiology and disease is highlighted by their impact on drug development, where NHRs are one of the main drug targets [1]. This is remarkable if one considers that there are only 48 NHRs in the human genome [2]. However, they represent more than 10% of all drug targets [1].

The NHR family has two hallmarks: they share homology to the steroid and thyroid hormone receptors [3] and their function is modulated by small molecules derived from endogenous metabolism or from the diet [4] (Figure 1). These features distinguish NHRs from other proteins that are modulated by metabolites, such as the nuclear receptors of the PAS family [5] and membrane receptors of the GPCR family [6,7]. Although much is known about the biology of NHRs and the ligands associated with them, many questions remain unanswered. Did NHRs evolve from metabolic sensors? What was the origin of the hormone function? What determines the specificity of an NHR for its ligands? Can NHRs be regulated by more than one ligand? Do NHRs regulate ligand biosynthesis through feedback loops? The identification of physiological ligands for NHRs is central for answering these important mechanistic questions and is of great interest as a source of drug targets for the treatment of many diseases including cancer, metabolic syndrome and autoimmune and endocrine dysfunction.

NHR ligands include both synthetic and natural compounds that can act as receptor agonists or antagonists. We define physiologic NHR ligands as the natural metabolites that are present in the relevant tissues at half of the maximum effective concentration binding the receptor. When a ligand binds to the NHR ligand-binding pocket (LBP), the resulting conformational change in the NHR induces the recruitment of coactivators that promote transcription of the target gene [11]. Ligands that bind to the NHR LBP are orthosteric ligands [27] (Figure 2), and the majority of known physiologic NHR ligands belong to this group. Allosteric ligands that bind outside the LBP to regulate NHR transcription have been identified [27] (Figure 2), but most allosteric NHR ligands are synthetic compounds, which are part of a broader class of nuclear receptor alternate-site modulators (NRAMs) [28] and are outside the scope of this review. Examples of recent discoveries of synthetic NHR ligands with promising therapeutic potential include modulators for RORγ (NR1F3) [29], PXR (NR1I2) [30] and RXR (subgroup NR2B) [31].

The overall goal of this review is to provide an account of the relationship of NHRs and their physiologic ligands. This includes the co-evolution of NHRs and their ligands as well as the mechanisms that determine ligand-receptor specificity. Also included is a discussion of how these receptor–ligand interactions are co-regulated through one-ligand and dual-ligand recognition models and feedback loops. We conclude our review with a discussion on how physiologic ligands for NHRs can be identified.

## 2. The Origins and Functions of NHRs and Their Ligands

NHRs are found only in the animal kingdom (Figure 3), and no NHR homologues have been identified in archaea, bacteria, plants, protozoa or yeast. However, the transcriptional machinery used by NHRs was present in early eukaryotes. Indeed, the estrogen and glucocorticoid receptors can promote ligand-dependent transcription of synthetic NHR reporters in yeast [39,40] and plants [41], which naturally do not express NHRs. The number of distinct NHRs and NHR subfamilies differs between species. Humans have 48 NHRs [8] while the fruit fly *Drosophila melanogaster* has 20 NHRs [42], as do most insects [43]. In contrast, the nematode *C. elegans* has only 1000 somatic cells but over 230 NHRs [44], which include homologues for the NR1, NR2, NR4, NR5 and NR6 subfamilies [45]. The large number of NHRs in nematodes is the result of the massive expansion of HNF4A (NR2A1) [46] and reflects a chemosensory adaptation that is unique to nematodes [47]. The hormones and pheromones that control the biology of these worms are derivatives of fatty acids such as ascarosides [48] and should be explored as potential ligands for these receptors.

The origin of NHRs can be traced back to yeasts, which have two transcription factors, Oaf1 and Pip2, which contain an NHR ligand-binding domain (LBD)-like fold that can bind the fatty acid oleate and regulate its metabolism [59]. This suggests that the NHR family may have evolved from ancestral proteins dedicated to the regulation of fatty acid metabolism in early eukaryotes. An alternative hypothesis is that the NHR fold is similar to terpene synthases [60], suggesting that the ancestral ligand for NHRs was a terpenoid. Indeed, most NHR ligands, like retinoids and sterol lipids [4], are terpenoids. Although Houston et al., 2022 [60] could not exclude the possibility that the structural similarities of NHRs and terpene synthases were due to convergent evolution, this hypothesis provides a competing view to the idea that the ancient NHR ligand was a fatty acid.

In agreement with the fatty acid hypothesis, most members of the NR2 subfamily recognize fatty acids and related metabolites. The members of the NR2A group bind saturated and monounsaturated fatty acids [50,51,52], the NR2C group binds polyunsaturated fatty acids (PUFAs) [61,62], NR2E1 (TLX) binds the fatty acid oleate [63], and the receptors NR2F1 and NR2F2 (COUP-TFs) bind sphingolipids [64]. Even RXRα (NR2B1) bind docosahexaenoic acid (DHA) which is a PUFA [65]. Thus, in pre-bilaterian animals, at least two families of ligands are recognized by NHRs: fatty acids, their derivatives and retinoids.

The RXR of placozoans and cnidarians bind 9-cis retinoic acid with a high affinity [55,56,66]. However, many bacteria, algae and invertebrates have the enzymatic machinery to produce PUFAs endogenously [67]. Thus, it is not clear if the physiologic RXR ligand in early metazoans is a retinoid or a fatty acid. In any case, the modifications required to convert a fatty acid binding receptor into a retinoic acid binding receptor are subtle, and there is a striking similarity between the RXR contact residues interacting with 9-cis retinoic acid [68] and DHA [69]. Other members of the NR2 subfamily have retained the ability to bind retinoids at supra-physiologic concentrations. These include NR2C2, NR2E1 and NR2F2 [70,71,72]. We hypothesize that the NR2 subfamily evolved to recognize relatively abundant metabolites with high specificity. This allowed these receptors to maintain weak binding to terpenoids, because the physiological concentration of terpenoids in tissues is below the levels required to compete with the physiologic ligand. This situation is analogous to that of the ancestral steroid receptor, which evolved in animals with a less diverse metabolome [73]. The lack of metabolic diversity reduced the selective pressure on the receptor for ligand specificity [73].

It seems that the metabolome exerts selective pressure on evolving NHRs. First, a ligand has to be present, and there is evidence that the ligand biosynthetic and catabolic machinery co-evolves with NHRs. This was observed for steroid hormones and their receptors [74]. Similarly, orthologs of the enzymes that process carotenoids and retinoic acid, such as the β-carotene cleavage enzymes (BCOs), ALDH1 and ALDH2, CYP26A1 and SDR16C are found at the root of the metazoan tree [75,76,77,78]. Second, metabolic innovations may generate new ligands that reduce the selective pressure on existing receptors, allowing for NHR expansion. A good example of this strategy is the steroid hormone receptors, which evolved from an ancestral ER. This ancestor bound only estrogen-like compounds [79] and still had a cholesterol side-chain (Figure 4A) such as paraestrol A [80] (Figure 4B). A metabolic innovation occurred in vertebrates with the removal of the side-chain of cholesterol by CYP11A1 [81]. This is one of the first steps in steroid hormone biosynthesis [82] and it may have provided the starting point for an expansion of NHRs that recognize a diverse repertoire of steroids lacking a side-chain [80] (Figure 4B). Finally, the ligand itself provides selective pressure on the receptor to increase or decrease its specificity for a ligand. This phenomenon is observed in the interaction of bile acids and FXR (NR1H4) (Figure 4C). Bile acids are metabolites unique to vertebrates [83] and come in two forms: the bile acids of fishes and lampreys, which have 27 carbons and a flat sterol A-ring backbone, and most of the bile acids of mammals, which normally have 24 carbons and a bent sterol A-ring [83]. The bile acids of lampreys and fishes do not activate mammalian FXR [83,84]. In fact, the FXR of lampreys and fishes is well adapted to recognize bile acids with a flat A-ring, while mammalian FXR is well adapted to recognize bile acids with a bent A-ring [83,84].

By the time the first bilaterians evolved [85], the NHR family had expanded to contain all of the members of the NR2 subfamily as well as at least one ancestral receptor of the NR1, NR3, NR4, NR5 and NR6 subfamilies (Figure 3). The ability to bind sterols and porphyrins emerged in the urbilaterian ancestor. Thus, sterols are found as ligands for ecdysone receptors and Daf-12 in insects and worms (protostomes) [86,87] while oxysterols, bile acids, vitamin D and steroid hormones are sterol-like ligands for many vertebrate (deuterostomes) NHRs [4,74,78,82,88]. Porphyrins are ligands of the E75 NHR in *Drosophila melanogaster* [89] and for NR1D1 (REV-ERBα) and NR1D2 (REV-ERBβ) in vertebrates [90,91].

The evolution of the receptor–ligand interaction can provide us with a glimpse of the functional evolution of NHRs. If NHRs originated from a yeast ancestor that recognized fatty acids, it would suggest that the first function of NHRs was to regulate transcription in response to changes in membrane biosynthesis and cellular metabolism. How this cell-autonomous function has evolved into a hormonal function that has the ability to respond to signals produced by distal tissues, is still unclear. The simplest hypothesis would be that the hormonal function in NHRs originated from metabolic intermediates that could accumulate in tissues and act in an autocrine and paracrine fashion [4]. Over time, these compounds evolved to become dedicated hormonal metabolites. The finding that NR2F1 and NR2F2 bind 1-deoxysphingosines [64] suggests that the hormone function was present in non-bilaterian clades. 1-deoxysphingosines are non-canonical sphingolipids that are end-products of the sphingolipid biosynthetic pathway and cannot be converted into other sphingolipids such as phospho- or glycosphingolipids [92]. As such, they may not have metabolic roles. Wang et al. 2021, found that 1-deoxysphingolipids could activate NR2F1 and NR2F2 in an autocrine as well as a paracrine fashion [64]. Thus, the jury is still out as to whether NR2F1 and NR2F2 are receptors involved in the regulation of endogenous cellular metabolism or hormone receptors. Nevertheless, these findings suggest that a possible hormone-like function may have existed very early in non-bilaterian animals. Alternatively, the hormonal function may have evolved from a vitamin receptor such as RXR. Indeed, vitamins are unique metabolites dedicated to signaling. They have no other metabolic functions. Taken together, hormone-like compounds may have existed before the split from the urbilaterian ancestor.

## 3. Ligand-Receptor Interactions: Determinants of Specificity

Given that NHRs regulate key aspects of development and cellular physiology, their transcriptional activity has to be tightly regulated, as disruptions of this process can have serious consequences. There are mechanisms in place to ensure that NHRs are activated in the right place at the right time. One way to control receptor activity is to express the NHR at specific developmental times and cell types. Indeed, NHRs such as NR2E1 and FXR are quite restricted in tissue expression. NR2E1 is mainly expressed in the eye and brain, while FXR is expressed in the gut and urinary system (Human Protein Atlas https://www.proteinatlas.org/, (accessed on 24 July 2024). In this scenario, the ligands can be broadly distributed and abundant as receptor activation will only occur in the target cells. The second way to control receptor activation is by regulating ligand availability. The best example of this mode of regulation is the cycle of production and degradation of retinoic acid, which is essential for the development of the mouse hindbrain [93,94]. The enzymes that degrade retinoic acid form an anterior-posterior hindbrain gradient, while retinoic acid is produced by Raldh2 in the posterior mesoderm [93,94]. This generates a retinoic acid gradient from the high (posterior) to the low (anterior) region, and disruption of this gradient can severely affect the patterning of the mouse hindbrain [93,94].

Regulation of ligand availability is only meaningful if an NHR requires a ligand to be activated. NHRs have a common LBD structure with 12 α-helices surrounding a hydrophobic LBP core that varies in size from 0 to 1500 Å^3^ [15]. The crystal structures of several “empty” NHRs, such as NR4A2 (NURR1), NR0B1 (DAX1), NR0B2 (SHP1) and NR1D2, show LBPs that are too small to accommodate a ligand [95,96,97,98]. Thus, these NHRs were initially assumed to be ligand independent. However, the NMR-based identification of a canonical LBP in NR4A2 [99], the later discovery of prostaglandin A1 and DHA as potential physiologic NR4A2 ligands [100,101] and heme as a ligand for REV-ERBα and β [90,91] suggest that the small LBPs from crystal structures are artifacts of crystallography. Thus, all NHRs could be ligand-dependent.

NHR specificity is determined by the position and orientation of ligand-receptor contact residues, contact residue polarity and the shape of the LBP [15]. Steroid hormones have a hydrophobic core and two hydrophilic groups at opposite ends. The hydrophobic core of the ligand forms van der Waals interactions with the hydrophobic lining of the LBP whereas the hydrophilic groups in the ligand are tethered by hydrogen bonds to the opposite ends of the LBP [15]. This generalized binding mode applies to most NHRs, even when only one end of the ligand has a hydrophilic side, which is the case for fatty acids, retinoids or cholesterol. The first NHRs to be cloned, such as the glucocorticoid receptor [102], estrogen receptor [103], and retinoic acid receptors [104,105], had high affinities for their ligands [104,105,106,107]. This favored a view that an NHR–ligand interaction would follow a “lock and key” model similar to the one proposed by Emil Fisher to explain the specificity of enzymatic activity [108]. In this model, the ligand and receptor are rigid structures that bind to each other like a lock and a key. The “lock and key” model predicts that each receptor would recognize a unique ligand that interacts with the NHR LBP, which is static in volume and shape. A near perfect example of such a model is the interaction of the androgen receptor with testosterone (LBP = 584 Å^3^) and dihydrotestosterone (LBP = 582 Å^3^) [109]. However, other ligand-receptor interactions do not follow this simple model. Rather, many NHR–ligand interactions follow the “induced fit” model [110], which suggests that both the ligand and the LBP can undergo large conformational changes upon binding. Sometimes, the receptor is relatively static while the ligand adjusts its fit to the receptor. RORβ (NR1F2) binds and co-crystallizes with stearic acid [111] and all-trans-retinoic acid [112], two structurally different ligands. Likewise, RXR co-crystalized with both 9-cis retinoic acid [68] and DHA [69]. Remarkably, in each of these examples, the dissimilar ligands occupy the same areas of the LBP and share common contact residues with the receptor [69,109,111,112]. In other cases, however, the receptor undergoes large conformational changes to accommodate ligands. Such is the case for the glucocorticoid receptor bound to dexamethasone (LBP = 540 Å^3^) and deacylcortivazol (LBP = 1070 Å^3^) [113].

Not all NHR ligands bind to their receptors with a high affinity. FXR, for example, binds bile acids with a low affinity (50 μM) [114,115], and it also binds farnesol with a low affinity [116]. The correlation between ligand affinity and its tissue distribution and concentration defines the physiologic ligand. Indeed, the physiological concentrations of bile acids in the liver and gallbladder, where FXR is expressed, are 50–100 μM [117]. The ligand tissue concentration rule also applies to high affinity NHRs such as RAR-retinoic acid, whose concentration in mouse serum and tissues is in the nanomolar range [57]. In vitro-based assays often find ligands that are not physiological. Examples include the interaction of RXR and branched chain fatty acids [118], fatty acids and carotenoid metabolites [31] or NR2C, NR2E and NR2F subfamilies with retinoids [70,71,72]. These ligands bind the receptor with a low affinity and are present in tissue at low concentrations, which excludes them as physiologic ligands. The affinity–tissue concentration rule suggests that some NHRs are activated by different ligands in different tissues. For example, RXR transcriptional activity could be regulated by 9-cis-13,14-dihydroRA in the liver and blood [57] and DHA in the brain, where its concentrations are high enough to activate the receptor [65]. Similarly, NR2E1 transcriptional activity may be regulated by retinoids in the retina [72] and other ligands in the brain.

Another determinant of specificity in the NHR–ligand interactions is uniqueness. Hormones and vitamins such as steroid hormones, vitamin D, 1-deoxysphingosines or retinoic acid have a low similarity to other metabolites and undergo limited processing by the organism. Retinoic acid is metabolized into a limited set of hydroxy and keto derivatives [119] that play only a minor role in mouse development [120]. Structural uniqueness helps to reduce noise from unrelated metabolites, with estrogens and phytoestrogens being a case in point. Estrogen is derived from cholesterol and has a unique structure and limited metabolic variation. Phytoestrogens are amino acid derivatives produced by plants of the order Fabales (roses, beans and peas) and are natural estrogen receptor disruptors. Most dietary phytoestrogens bind to the estrogen receptor with affinities that are 100–1000 times lower than the natural estrogen receptor ligand estradiol, and they are not abundant enough to reach physiological tissue concentrations [121]. Structural uniqueness and the correlation between affinity and tissue abundance are more important than affinity alone when defining a physiological NHR ligand.

Finally, NHR–ligand specificity could be increased by dual-recognition systems, in which NHRs are regulated by two ligands. We propose four categories (A–D) of such dual-recognition systems for NHRs. The most studied dual-recognition events, which we call type A systems, use the formation of heterodimers between two NHRs that require their respective ligands to function. A good example of type A systems are the class II NHRs (Figure 1F) such as NR2F1 and NR2F2, NR4A1, PPARs, THRA and B (NR1A1 and NR1A2), RAR and VDR that form heterodimers with the promiscuous partner RXR (Figure 1B,F) [11]. The constellation of all possible type A regulated NHR heterodimers could be quite large [11,122,123,124]. A second category is type B systems, in which the specificity of the NHR is regulated by a transporter. An example of a type B system is PPARδ (NR1C2), whose transcriptional activity is promoted by PUFAs such as arachidonic acid and linoleic acid [125]. PPARδ also binds retinoic acid with a high affinity [126]. The interaction of PPARδ with retinoic acid is facilitated by fatty acid binding protein 5 (FABP5) [127], which confers tissue specificity for the PPARδ-retinoic acid interaction [127]. For type C systems, we propose that allosteric and orthosteric ligands cooperate to promote NHR function. Examples of type C systems are the cooperative effects of allosteric vitamin E metabolites with orthosteric ligand pioglitazone in promoting transcription by PPARγ [34]. One candidate for a type B or C system is RORγ. Its known orthostheric ligands are sterol-lipids [128,129,130,131,132]. A second RORγ ligand, 1-oleoyl-lysophosphatidylethanolamine, is active on T cells [133]. It remains to be determined whether 1-oleoyl-lysophosphatidylethanolamine is an orthostheric or allosteric RORγ ligand. Finally, we propose a hypothetical type D system, in which two ligand-dependent proteins, one of which is an NHR and the other of which is from a different protein family, form heterodimers to modulate the expression of a target gene. No example of a type D system has been identified as of yet.

## 4. Feedback Loops of Ligand Biosynthesis–NHR Regulation

A prominent feature of many NHRs is the presence of feedback loops in which the receptor directly regulates the synthesis and catabolism of its ligand (Figure 5). Such feedback loops are not always obvious; many NHRs, such as the thyroid hormone receptors [134] and retinoic acid receptors [135,136], have pleiotropic effects that affect many aspects of cellular metabolism.

A direct feedback loop is one in which the receptor directly regulates the expression of enzymes involved in the metabolism or catabolism of the ligand (Figure 5). Two classic examples of such direct feedback loops are the vitamin D receptor (VDR) (Figure 5) and the thyroid hormone receptor. The availability of thyroid hormone is controlled by the three deiodinases DIO1, DIO2 and DIO3 [134]. DIO1 and DIO2 convert the thyroid hormone precursors Tetraiodothyronine (T4) into the active thyroid hormone, Triiodothyronine (T3) [134], whereas DIO3 inactivates T4 by converting it into an inactive rT3 metabolite [134]. The levels of T4 and T3 are regulated by the thyroid hormone receptors TRHA andTRHB [143,144]. THRA upregulates DIO1 [145], increasing the conversion of T4 into T3. In contrast, THRB bound to T3 inhibits the transcription of DIO2 [146], thus reducing the conversion of T4 into T3. Furthermore, increased levels of T3 causes THRA to upregulate DIO3, reducing the levels of T3 and downregulating THRA transcriptional activity [147]. Other examples of direct feedback loops have been described for LXRα and β (NR1H3 and NR1H2), in which activation by oxysterols [148] leads to expression of CYP7A1, which promotes the catabolism of cholesterol and oxysterols into bile acids [149,150]. Similarly, activation of RAR by retinoic acid leads to the expression of the enzyme that degrades retinoic acid (CYP26A1) by retinoic acid receptors [151].

However, the direct feedback loops described above may be the exception rather than the rule. Some NHRs have indirect feedback loops. FXR inhibits the synthesis of bile acids through an indirect loop, increasing the expression of NR0B2, which then interacts with NR5A2 (LRH1) to shut down CYP7A1, the first step in bile acid biosynthesis [149,152,153]. More complex patterns of feedback regulation emerge as NHRs and their ligands are linked to one another. Oxysterols and bile acids are part of cholesterol catabolism. Accordingly, LXR and FXR are intertwined, with one (LXR) promoting bile acid synthesis and the other (FXR) inhibiting bile acid synthesis [152].

Finally, there is a third group of NHRs in which no clear feedback loop is present. This group is represented by the steroid hormone receptors. Steroid receptors, such as the estrogen, androgen, mineralocorticoid and glucocorticoid receptors, regulate the physiology of complex multiorgan systems linking the brain, pituitary gland and steroidogenic tissues into the circuity of steroid hormones. Steroids are products of cholesterol processing in which one biosynthetic intermediate can serve both as a ligand for one NHR and the ligand precursor of another NHR [4,82]. Despite this complexity, steroid hormone receptors, with the exception of the progesterone receptor [154], do control the levels of steroid hormones. There is a 10-fold increase in the level of serum estradiol in estrogen receptor knockout mice [155]. Mutations in the androgen receptor can cause testicular feminization which produces females with the X,Y karyotype in mice [156], rats [157] and humans [158,159]. In mice with testicular feminization, the levels of testosterone, the androgen receptor ligand, are reduced due to low expression of the steroidogenic enzyme Cyp17a1 [160,161]. Similarly, glucocorticoid receptor knockout mice have increased levels of corticosterone [162] and mineralocorticoid receptor knockout mice have a 60-fold increase in aldosterone, the mineralocorticoid receptor ligand [163]. However, in all of these cases the evidence that the steroid hormone receptors directly regulate the hormone biosynthetic enzymes is either weak or absent. Surprisingly, the main regulators of steroidogenesis are NR5A1 (SF1) and NR5A2. These receptors regulate cholesterol biosynthesis, uptake and transport as well as the mitochondrial cholesterol metabolism required for steroidogenesis [164]. Furthermore, NR5A1 and NR5A2 bind to and enhance transcription of key steroidogenic enzymes and transporters in the steroid hormone biosynthetic pathway [164]. To complicate matters, the ligands for NR5A1 and NR5A2 are phosphatidylinositols [165], which are neither structurally nor biosynthetically related to sterols. At this point, there are no studies that establish a strong connection between steroid hormone receptors and the regulation of phosphatidylinositol biosynthesis or the expression of NR5A1 and NR5A2.

Feedback loops are typically identified after the ligand itself has been found. The presence or absence of direct or indirect receptor–ligand feedback loops in recently deorphanized receptors needs to be determined. The power of identifying such loops is illustrated by RORγ. Cholesterol biosynthetic intermediates are orthosteric ligands for RORγ [128,132]. Cai et al., 2019, showed that RORγ cooperates with SREBP2 to regulate cholesterol biosynthesis [166]. However, in contrast to the previous examples of direct feedback loops presented here, where the receptor inhibits ligand synthesis, RORγ promotes cholesterol biosynthesis [166]. This provides a positive feedback loop that can be targeted for cancer therapy [166,167,168]. Thus, feedback loops themselves can become therapeutic targets. Furthermore, finding a feedback loop is a good indicator that a receptor has been deorphanized. The finding of 1-deoxysphingosine as the ligand for NR2F1 and NR2F2 suggests a feedback loop for these receptors [64]. The production of sphingolipids is modulated by the availability of palmitic acid [169] and the relative balance of alanine and serine in cells [169,170]. The expression of CPT1A, the enzyme responsible for the oxidation of palmitic acid, is reduced in NR2F1 and NR2F2 double deficient cells [64]. The induction of 1-deoxysphingosine biosynthesis in WT cells decreases the production of palmitic acid while increasing the production of palmitoyl-carnitine, the first intermediate in the oxidation of palmitic acid [64]. Similar effects are observed in adipocytes treated with 1-deoxysphingosines [171]. Since palmitate is the precursor of 1-deoxysphingosines, NR2F1 and NR2F2 may be providing a negative feedback loop by simultaneously reducing the synthesis of the ligand precursor (palmitate) and increasing its degradation (oxidation). Further studies are needed to establish whether these effects are due to a direct feedback loop between NR2F1/2 and sphingolipid biosynthesis.

## 5. Identification of NHR Ligands and Limitations of Present Approaches

The concept of hormones was revolutionary. It stipulated that an organ produces a metabolite that is released into the blood to regulate another organ [4], and thus provided a hypothesis-driven framework for the identification of hormones. A biological observation became a bioassay, and one could search for the hormone by isolating compounds from tissues and testing their bioactivity. The thyroid hormone was identified by its ability to treat hypothyroidism in sick animals and humans or cause hyperthyroidism in healthy animals [172,173]. Thus, by identifying highly active fractions in thyroid extracts, the thyroid hormone was isolated and its structure determined [173]. However, the proof of the pudding was the synthesis of the bioactive thyroid hormone in the lab [174]. This procedure also applies to vitamins. A lack of vitamin A causes xerophthalmia and blindness in animals [175]. The animals can be cured by fish liver oil, which can be fractionated until one has pure vitamin A for structure determination [176] and synthesis [177]. A similar framework was used to solve the mystery of metamorphosis (Figure 6).

The discovery of hormones and how they were connected with their ligands was previously discussed [4]. The current framework suggests that the problem for NHR ligand identification is similar to that of hormone identification. The advantage we have is that by knowing the receptor, one can replace the complex bioassays used to identify hormones and vitamins by using cell-based assays to measure the effect of a ligand on NHR transcriptional activity. These assays use cells transfected with full-length or chimeric NHR proteins [187,188,189,190] plus a reporter [191,192,193]. Ligands are identified using procedures similar to those used to identify classic hormones, such as using metabolites fractionated from tissues [100], combinations of biochemical and candidate approaches [57,87,194,195,196], combinations of genetic, enzyme overexpression screens and candidate metabolites [64,128] or candidate approaches where metabolites are tested individually [148,197,198].

The ligands for NHRs are produced either by endogenous metabolism or are derived from the diet. The set of biosynthetic pathways and metabolites that form the endogenous metabolome is relatively conserved between species [199] and contains a few thousand molecules [200]. The diversity of the endogenous metabolome can be increased by modifications of endogenous metabolites by the microbiome. One example is secondary bile acids, which are microbiome metabolites derived from the processing of primary bile acids, which are endogenous metabolites for vertebrates. Secondary bile acids bind and modulate the activity of FXR [201] and RORγ [202]. Furthermore, the microbiome may generate metabolites such as indole-3-propionic acid, which are weak agonists for PXR [203]. However, the collection of all possible exogenous metabolites in nature is very large and contains over 3000 chemical classes and millions of molecular entities [204]. Only a small fraction of this chemical universe is of interest where physiological NHR ligands are concerned. Most NHR families are widespread throughout the animal kingdom, which suggests that their ligands are derived from common metabolites and dietary compounds such as sterols [205,206], vitamin A [207] and PUFAs [67]. While these metabolites may be common, there are still thousands of them. Therefore, it is important to reduce the complexity of the task by identifying the ligand precursor or its biosynthetic pathway.

Ligand identification systems depend on sensitive cell-based assays, which have to be designed to detect transcriptional activation or repression by the NHR. Experimental evidence suggests that most NHRs function both as activators and as repressors of transcription, depending on the genomic context. For example, in the absence of a ligand, the thyroid hormone receptor represses transcription through the recruitment of a co-repressor [208,209], while the addition of thyroid hormone promotes transcriptional activation [210]. Most cell-based assays are designed to identify transcriptional activation and ligand agonist activity, although these assays could be modified to detect antagonists, as exemplified by the identification of digoxin as a RORγ antagonist [211]. Another problem with cell-based assays is that some ligands may not activate or repress NHR transcriptional activity in these assays. This may just reflect the fact that we do not know enough of the biology of some NHRs to define how to measure their receptor activity. The presence of endogenous agonists in the cell-based assay can hamper the search for physiologic ligands. This can be overcome by the development of an agonist-free cell-based assay similar to the one employed to show that RORγ transcriptional activity is dependent on sterol lipids [128]. Ligand-free cell-based assays could be used to develop assays that detect weak agonists. Furthermore, ligand-free systems can be used to titrate the concentration of known agonist to develop screens for weak antagonists and inverse agonists. Although desirable, it is not always possible to generate ligand-free systems. Certain metabolites, for example sphingolipids, are essential elements of eukaryotic cell membranes [212,213]. Thus, the ligands for NR2F1 and NR2F2 had to be identified in ligand-sufficient systems [64].

A ligand’s biosynthetic pathway can be identified if the cell line used for the assays is producing the ligand endogenously in small amounts. This allows for overexpression screens of metabolic enzymes in bioactive cells. The idea of these overexpression screens is to enhance ligand biosynthesis, which is detected using a cell-based reporter [64,128]. Thus, the first step is to determine whether the ligand is produced by the cells. For example, insect cells are auxotroph for sterol-lipids, and RORγ transcriptional activity is completely inhibited in insect cells grown in a sterol-free synthetic medium [128]. This is useful for screens designed to identify variations of sterol-based agonists to build structure activity relationships [128]. However, enzyme overexpression screens for RORγ ligands cannot be performed in insect cells, because they lack an endogenous sterol biosynthetic pathway to be amplified. Similar care has to be taken with *C. elegans,* whose development is entirely dependent on the presence of contaminating cholesterol in the agar plates used to grow the worms [214].

One may argue against using a metabolic enzyme overexpression screen if one can add the metabolite directly to cells. However, direct metabolite screens could be affected by multiple variables. One of them is whether cells can take up the ligand or its precursor added to the culture medium. The transcriptional activity of FXR to conjugated bile acids is only detected when FXR is co-transfected with the bile acid transporter SLC10A2/IBAT [115]. Similarly, 1-deoxysphingolipids need to be conjugated to BSA to be taken up by cells and induce NR2F1 or NR2F2 transcriptional activity [64]. Enzyme overexpression screens could bypass these limitations, because they affect endogenous ligand synthesis and are independent of the ability of cells to take up ligand or precursor. Furthermore, once the ligand biosynthetic pathway has been identified, it can be manipulated to generate sufficient ligand for biochemical characterization to define its structure and synthesis. As our knowledge of the metabolome and ligand biosynthetic pathways advances, a complete set of synthetic metabolites and cell delivery techniques to perform direct ligand screens will eventually be developed.

The induction of reporters in cell-based assays can be indirect, and the ligands identified in these assays need to be confirmed as directly binding to a receptor. Early studies used very sensitive assays employing a radiolabeled ligand on cell extracts of mammalian cells transfected with glucocorticoid [102], estrogen [103] and retinoic acid receptors [104,105]. The production of recombinant protein from bacteria allowed one to measure NHR–ligand interactions using non-radioactive methods such as fluorescence polarization [215], surface plasmon resonance [216] and NMR [217]. Direct binding to NHR is a strong indication that a metabolite is a ligand. However, as direct screening methods, they have their own limitations. Each ligand class has differences in solubility and chemical properties, such as the formation of micelles in the buffers used for the assays. This requires ligand binding assays to be adjusted for the class of ligands being investigated and to be specific for fatty acyl amino acids [218], sphingolipids [169] or other lipid groups. Direct binding assays also increase the rate of false-positives, and ligands detected using biophysical methods need to be confirmed by cell-based assays, binding affinity, tissue concentration and function “in vivo”.

Ligand identification is a “work in progress” process. The identification of a high affinity ligand defines the class and subclass of ligands for a given NHR in the natural chemical universe. This facilitates the discovery of other bioactive metabolites. The first androgen isolated as the male hormone was androsterone, which was isolated from urine (3α-hydroxy-5α-androstan-17-one) [219]. Urine has a very small metabolome [220], which facilitates isolation of its compounds. However, knowing the chemical characteristics of androsterone facilitated the isolation of similar compounds directly from testicles, leading to the identification of testosterone (androst-4-en-17b-ol-3-one) years later [221]. Similarly, the first identified metabolites of vitamin D [222,223] had little biological activity [224,225]. Indeed, the activation of vitamin D requires the hydroxylation of pro-vitamin D to form 25-hydroxyvitamin D3 [226] and then 1α,25-dihydroxyvitamin D [227]. Finally, the comparative metabolomics of dafachronic acids have identified new endogenous, tissue-specific members of the dafachronic acid group, which are ligands for DAF-12 [228]. Similar findings will occur as NHRs are deorphanized.

## 6. Conclusions

NHRs are a family of transcription factors that play an important role in the regulation of many aspects of development and physiology in animals. NHRs are important for medicine, pest control and farming. Most of our knowledge so far is based on the functions of orthosteric NHR ligands. In this review, we put forward models describing the ancestral NHR ligand and the origin of the hormone function, the mechanisms that increase ligand specificity, the affinity–tissue concentration rule, dual-recognition systems and how to damp the effect of the “noisy background” caused by other metabolites. We explored the role of ligand biosynthesis–NHR regulatory feedback loops. This, in spite the fact that physiological ligands for many NHRs are still unknown. Finding these ligands is paramount for our understanding of NHR biology and function. We predict that new NHR ligands will be derived from common metabolites that are widespread in nature. The emerging class of allosteric ligands may enter center stage and become as important as orthosteric ligands. Defining the contribution of allosteric and orthosteric ligands to NHR activity may require a new generation of cell-based assays. Ligand identification is a “work in progress”, in which the identification of one high affinity ligand gives us the class and subclass in the natural chemical universe that binds the receptor and facilitates the discovery of other bioactive metabolites. Today’s resources would help deorphanize most NHRs in a decade or two. This would greatly advance our understanding of the development and physiology of the animal kingdom and help treat human diseases. 

## Figures and Tables

**Figure 1 cells-13-01284-f001:**
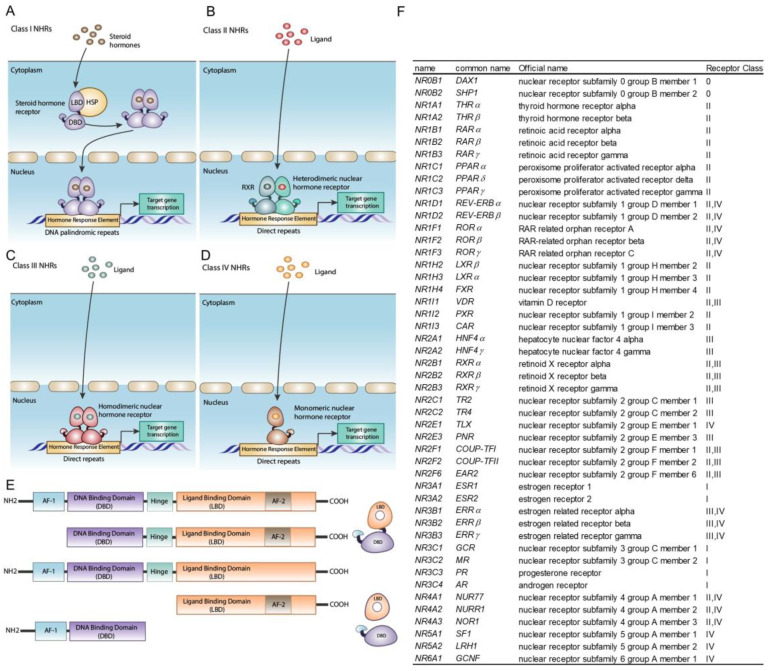
NHRs are classified using a standardized nomenclature [8]. Functionally, the 7 subfamilies of NHRs in the human genome can be grouped into 4 classes [9]: (**A**) Class I NHRs are ligand induced receptors that bind to DNA palindromic repeats as homodimers such as the steroid hormone receptors [9,10]. (**B**) Class II NHRs form heterodimers with the members of the NR2B group (RXRs) and bind to direct repeats in the DNA sequence [9,10]. Some examples of Class II NHRs are the members of the NR1B group (RAR) and NR1I1 (VDR) [11]. (**C**) Class III receptors form homodimers that bind to direct repeats in the DNA [9]. An example of Class III NHR are COUP-TFs (NR2F1 and NR2F2) [12]. (**D**) Class IV NHRs bind to DNA extended core sites as monomers [9]. NR4A1 (NUR77) and NR5A1 (SF-1) are Class IV NHRs [13,14]. (**E**) NHRs of all classes have a modular structure with 4 domains: Two transactivation domains, activation function (AF-1) at the N-terminus and AF-2 at the C-terminus [15] and between AF-1 and AF-2 there is a DBD and LBD [15]. The modular structure is not rigid and variations in the basic design are common. For example, the estrogen receptor (NR3A1) has isoforms that lack the AF-1 domain [16] and the glucocorticoid receptor (NR3C1) has isoforms that lack the AF-2 domain [17]. The loss or truncation of the AF-1 domain in steroid receptors can result in tissue-specific patterns of ligand action [16]. The loss of the AF-2 domain affects both ligand binding and transcriptional activity generating isoforms that are transcriptionally repressive [17]. Furthermore, there are isoforms of the androgen receptor (NR3C4) that lack an LBD domain [18]. Finally, some NHRs that function as transcriptional repressors lack the DBD domain. These include NR0B1 (DAX1) and NR0B2 (SHP1) and isoforms of NR2F2 (COUP-TFII) [19,20,21]. (**F**) NHR nomenclature and class distribution. The table contains the standard NHR nomenclature [8], the common name, the official gene symbol defined by the HUGO gene nomenclature committee and the official gene name. The last column shows the classification of each receptor by class I to IV (panels (**A**–**D**)), which is based on references [22,23,24,25,26].

**Figure 2 cells-13-01284-f002:**
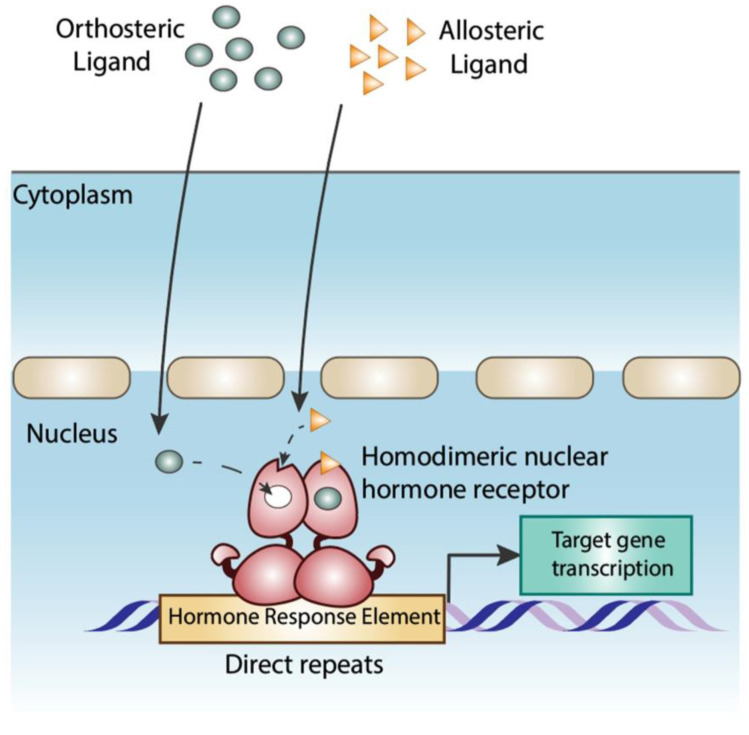
Orthosteric and Allosteric NHR ligands. Orthosteric ligands bind inside the NHR LBP while allosteric ligands bind on alternative sites in the LBD. Most known physiologic ligands for NHRs, such as retinoic acid and steroid hormones, are orthosteric. In contrast, the concept of physiologic allosteric ligands for NHRs is relatively novel [32,33] and understudied. However, they add an important conceptual tool for understanding NHR transcriptional activity. Physiologic allosteric ligands are defined as those metabolites that regulate receptor activity by binding outside the NHR LBP. Allosteric ligands were first described for estrogen receptors. The oxysterols 24(S), 25 and 27-hydroxycholesterol are natural metabolites that bind and inhibit the transcriptional activity of the estrogen receptors ERα (NR3A1) and ERβ (NR3A2) at physiologic concentrations [32,33]. Other examples of natural allosteric ligands are metabolites of vitamin E (α-tocopherol) which bind to PPARα (NR1C1) and PPARγ (NR1C3) [34,35] in vitro, and thyroid hormone (T3), which binds to an allosteric site of the androgen receptor [36]. However, the low concentration of metabolites of α-tocopherol [37] and T3 [38] in serum suggests that these allosteric ligands are not physiologic.

**Figure 3 cells-13-01284-f003:**
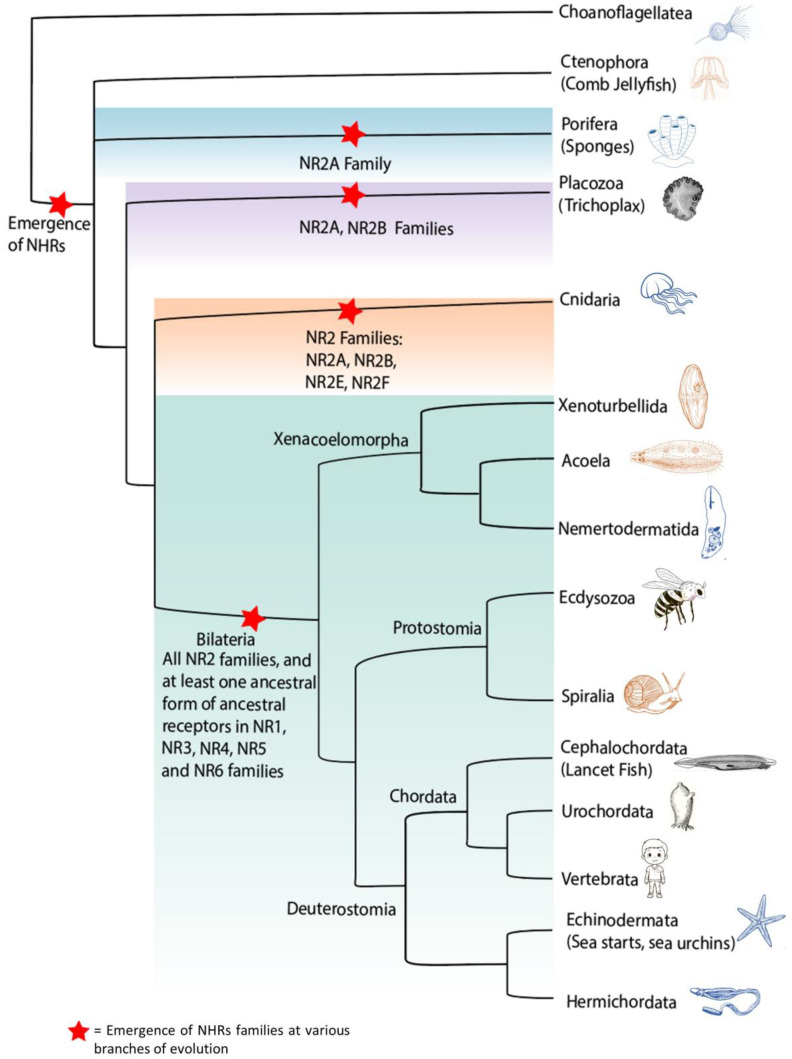
NHR families in the animal kingdom. The first authentic NHRs are found in sponges like the demosponge *Amphimedon queenslandica*. These NHRs recognize saturated and monounsaturated fatty acids and include an ortholog of the mammalian NR2A group [49], which are known receptors for saturated and monounsaturated fatty acids [50,51,52]. Next, the placozoan *Trichoplax adhaerens* has 4 NHRs, including orthologs for the NR2A, NR2B [53] and the NR2F groups [54]. The RXR of *Trichoplax adhaerens* recognizes a terpenoid [55,56,57], suggesting that the branching of fatty acid versus terpenoid ligands occurred early in evolution. In cnidarians such as *Nematostella vectensis*, there are 17 NHRs, including orthologs for all members of the NR2 subfamily as well as potential ancestors for the remaining NHR subfamilies [58].

**Figure 4 cells-13-01284-f004:**
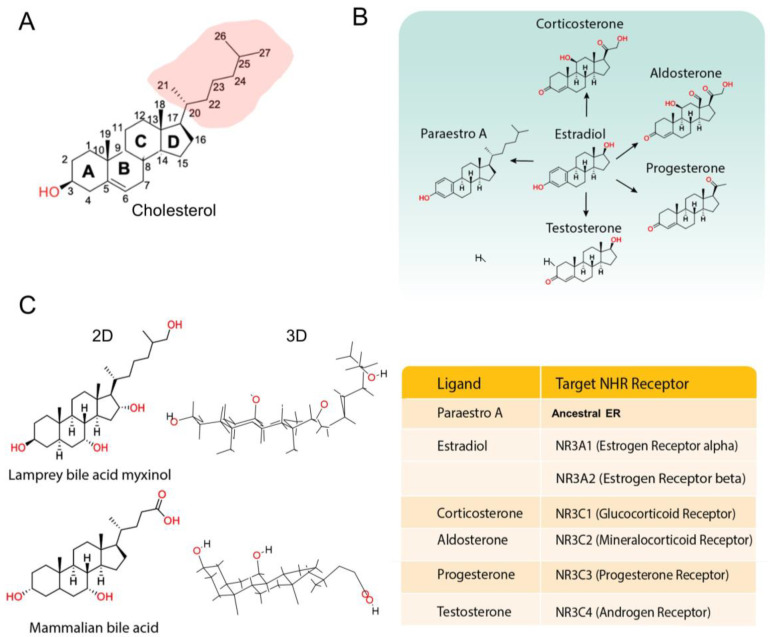
NHR–ligand co-evolution. Ligands exert selective pressure on NHRs. (**A**) Structure of cholesterol. (**B**) Evolution of steroid hormones from a cholesterol-like metabolite. (**C**) Adaptation of FXR to recognize bile acids by differences of 2D and 3D structures.

**Figure 5 cells-13-01284-f005:**
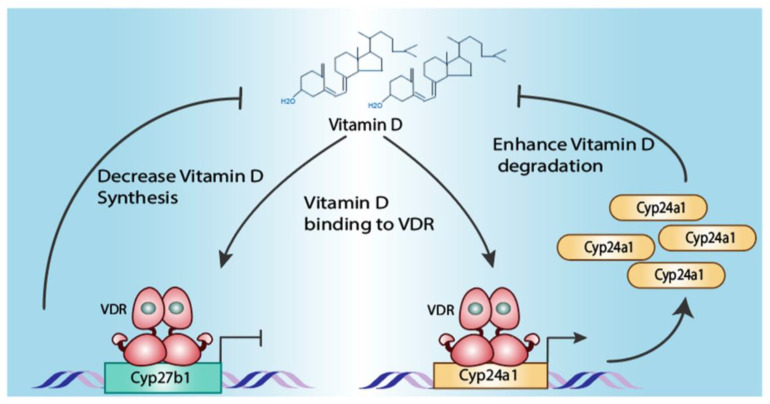
Homeostasis of Vitamin D action is regulated through a direct feedback loop. In the presence of vitamin D (1α,25-dihydroxyD3), the VDR binds to the promoters and enhancers of the enzymes that are responsible for the activation of vitamin D (CYP27B1) [137,138] and its catabolism (CYP24A1) [139,140,141]. VDR binding leads to the transcriptional repression of the vitamin D activating enzyme CYP27B1 [137,138,142], and transcriptional activation of the vitamin D inactivating enzyme CYP24A1 [139,140,141]. This is reflected by increased levels of vitamin D in VDR knockout mice caused by lack of both transcriptional repression of Cyp27b1 and transcriptional induction of Cyp24a1 [142].

**Figure 6 cells-13-01284-f006:**
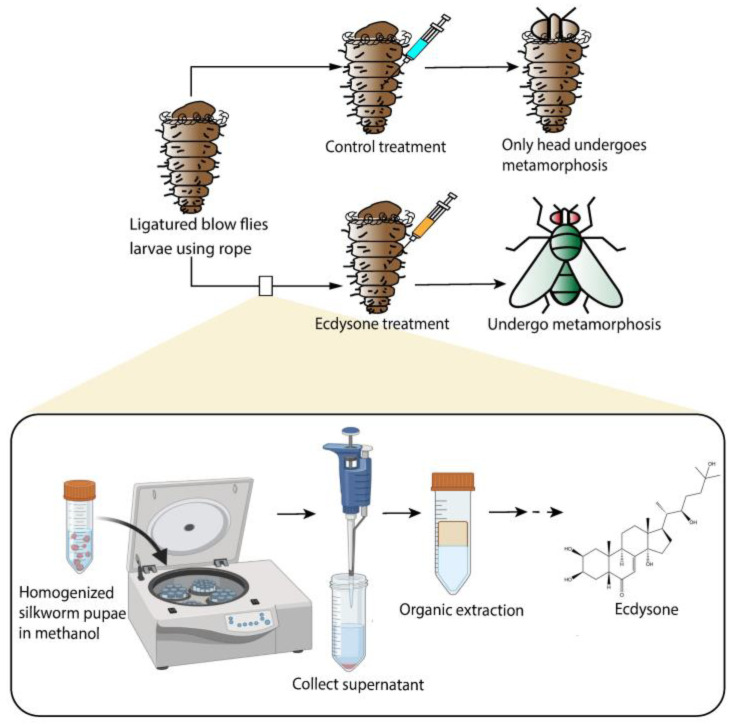
The isolation and identification of ecdysone as the hormone that regulates molting and metamorphosis in insects. Insects have a chitin-based exoskeleton and need to molt in order to grow. Molting in the kissing bug *Rhodnius prolixus* is inhibited by decapitation [178] and metamorphosis in butterflies and moths (*Lepidoptera*) is blocked by removing the brain in these animals [179]. This suggested that the head of insects produces a “hormone” that is released into the hemolymph to promote molting or metamorphosis [178]. To identify this hormone, a sensitive bioassay was developed in which a ligature was used to block the circulation of hemolymph from the head to the rest of the body of blow fly larvae [180]. Under these circumstances, the head will undergo metamorphosis while the rest of the body remains in the larval stage. However, these larvae will undergo metamorphosis when injected with insect extracts that contain the hormone [180]. In this manner, ecdysone was isolated [181], characterized [182,183] and synthesized [184,185,186].

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
