# Peer review of "Nuclear Receptors and the Hidden Language of the Metabolome"

_cells, 2024, doi:10.3390/cells13151284_

Round 1

Reviewer 1 Report (New Reviewer)

Comments and Suggestions for Authors

Tracking #: Cells - 3079975Yujie Chen et al.

The review article entitled “Nuclear receptors and the Hidden Language of the Metabolome” provides an update on the identification of physiologic ligands for nuclear receptors (NRs). The review also proposes models of ligand-NR co-evolution and of NR activity evolution.

Given the importance of the signalling pathways controlled by nuclear receptors and the therapeutic potential of these proteins, this review is interesting and original, and adequately documented with an extensive literature. It represents a major effort. However, this article is not entirely a review, and is more the authors' personal view of the evolution of NRs and their ligands, and seems to me to be relatively speculative. On the whole, the text is a bit tedious to read and I would like to see some improvement in the writing.

The abstract clearly announces what the reader will find in the paper. The figures are correct and sufficiently informative to illustrate the text.

The following is a list of few points that should be clarified and/or improved:

1/ Figure 1: the legend needs to be more precise and rewritten. Some NRs are mentioned by their nomenclature name, while others are mentioned by their use name. It is necessary to harmonize and indicate the nomenclature name at least once.

Various examples of particular modular organizations are also cited, but it is necessary to refer to the bibliography to find out which receiver is involved (e.g. p3 lines 48-50). Please specify which NR it is.

2/ For the reader's convenience and clarity, a table listing all human receptors and their names should be added. It would also be appreciated if this list could be classified according to the 4 NR classes shown in figure 1.

3/ line 55: NR ligands are specified as synthetic and natural compounds. Concerning synthetics, it would be desirable to specify that they can be agonists, like most natural molecules, or antagonists. It should also added that endocrine disruptors and molecules from the gut microbiota can be ligands for NRs.

4/ Figure 3 legend: Please note that the subject of the existence of an endogenous RXR ligand is still highly debated (line 120). 9cRA should not be considered as such (line 101). Also, using this compound as a molecule to study evolution is questionable, as its structure is very similar to fatty acids anyway, and indeed the 3D structure of RXR has been solved with all these compounds. Concerning DHA (line 123), its agonist activity is very weak, and we must be cautious about its possible physiological role (line 222).

5/ line 110: please clarify what NHR Ur ligand is.

6/ line 194: please note that, in addition to the position and orientation of ligand-receptor contact residues, the nature of the residue (polar or non-polar) and the shape of the LBP (which depends on the first two factors mentioned) determine the specificity of NRs.

7/ line 351: please add ligands provided by gut microbiota.

8/ line 374: the problem of detecting compounds that bind but exert weak activation or antagonistic activity is not mentioned. Also other methods (biophysical for example) should be considered to overcome the limitations and problems encountered with eukaryotic cellular models.

9/ Typo: spaces to be deleted (lines 34, 192, 200, 322, 354).  

Author Response

Reviewer 1 comments:

The review article entitled “Nuclear receptors and the Hidden Language of the Metabolome” provides an update on the identification of physiologic ligands for nuclear receptors (NRs). The review also proposes models of ligand-NR co-evolution and of NR activity evolution.

Given the importance of the signalling pathways controlled by nuclear receptors and the therapeutic potential of these proteins, this review is interesting and original, and adequately documented with an extensive literature. It represents a major effort. However, this article is not entirely a review, and is more the authors' personal view of the evolution of NRs and their ligands, and seems to me to be relatively speculative. On the whole, the text is a bit tedious to read and I would like to see some improvement in the writing.

The abstract clearly announces what the reader will find in the paper. The figures are correct and sufficiently informative to illustrate the text.

The following is a list of few points that should be clarified and/or improved:

Comment 1/ Figure 1: the legend needs to be more precise and rewritten. Some NRs are mentioned by their nomenclature name, while others are mentioned by their use name. It is necessary to harmonize and indicate the nomenclature name at least once.

Various examples of particular modular organizations are also cited, but it is necessary to refer to the bibliography to find out which receiver is involved (e.g. p3 lines 48-50). Please specify which NR it is.

Response: We modified extensively the legend in figure 1. We agreed with the reviewer that our naming of each receptor is generating confusion. This is also reflected in the literature where sometimes these receptors are defined as superfamily, families, subfamilies etc… To clarify this issue, we will use in most of the paper the standard name for each receptor. Except when the common name is so prevalent that using the standard name would create confusion. Nonetheless, we will add the standard name in brackets the first time the common name of a receptor is used. We hope this will clarify the issue. When referring to subfamilies and groups, we will use the official gene name. This addresses also a concern by reviewer #2 item 5. We also defined which NRs are involved in the modular organizations of p3 lines 48-50.

Comment 2/ For the reader's convenience and clarity, a table listing all human receptors and their names should be added. It would also be appreciated if this list could be classified according to the 4 NR classes shown in figure 1.

Response: We added a table with this information as panel F in Figure 1.

Comment 3/ line 55: NR ligands are specified as synthetic and natural compounds. Concerning synthetics, it would be desirable to specify that they can be agonists, like most natural molecules, or antagonists. It should also added that endocrine disruptors and molecules from the gut microbiota can be ligands for NRs.

Response: We added in now line 60 that synthetic and natural compounds can be both agonists and antagonists. However, we did not add endocrine disruptors and metabolites derived from the microbiota to this section. Since we claimed that “NHR ligands include both synthetic and natural compounds…” we thought there was no need to subdivide these compounds further. It would reduce the clarity of the text. We did add ligands produced by the microbiota on line 351 (now lines 360-364) as recommended by the reviewer on item 7.

Comment 4/ Figure 3 legend: Please note that the subject of the existence of an endogenous RXR ligand is still highly debated (line 120). 9cRA should not be considered as such (line 101). Also, using this compound as a molecule to study evolution is questionable, as its structure is very similar to fatty acids anyway, and indeed the 3D structure of RXR has been solved with all these compounds. Concerning DHA (line 123), its agonist activity is very weak, and we must be cautious about its possible physiological role (line 222).

Response: We agree with the reviewer that 9-cisRA is most likely not the physiologic ligand for RXR as we pointed out in line 120 (now 126-127) and also in line 222-223 (now 228-231) where we cite a manuscript suggesting that the real physiologic ligand for RXR is 9-cis-13,14-dihydroRA (ref 113). Therefore, we changed the text in line 107 “The RXR of Trichoplax adhaerens recognizes a terpenoid50-52 suggesting that the branching of fatty…” and added reference 113 (now 52) to the text. We also added “…DHA in the brain where its concentrations are high enough to activate the receptor60.” On line 223 (now line 230). We know that DHA binds RXR with lower affinity than 9-cisRA and is a weak agonist. The affinity-tissue concentration rule presented on lines 220-223 (now line 228-231) implied that DHA is not a RXR ligand in most tissues, except for the brain where its concentrations are high enough to activate the receptor.

We respectfully disagree with the reviewer on the similarity between DHA and 9-cisRA. These compounds are chemically very different. The Tanimoto coefficient of DHA and 9-cisRA is just 14.24%. They are also derived from different biosynthetic pathways and use different metabolic building blocks (9-cisRA is based on isoprene units) as we discussed in our previous manuscript (Tao et al 2020). Thus, studying how NHRs evolved to recognize these different types of ligands may suggest how metabolic pathways influenced the evolution of NHRs.

Comment 5/ line 110: please clarify what NHR Ur ligand is.

Response: we removed the term ur ligand and replaced it by “ancestral ligand for NHRs…” in now line 115. 

Comment 6/ line 194: please note that, in addition to the position and orientation of ligand-receptor contact residues, the nature of the residue (polar or non-polar) and the shape of the LBP (which depends on the first two factors mentioned) determine the specificity of NRs.

Response: Thank you for the comment. We changed it to: “…position and orientation of ligand-receptor contact residues, contact residue polarity and the shape of the LBP15.” These modifications are in new lines 200-201.

Comment 7/ line 351: please add ligands provided by gut microbiota.

Response: This was a great suggestion. We added this text to the manuscript in lines 362-366:

"The diversity of the endogenous metabolome can be increased by modifications of endogenous metabolites by the microbiome. One example are secondary bile acids which are microbiome metabolites derived from the processing of primary bile acids which are endogenous metabolites for vertebrates. Secondary bile acids bind and modulate the activity on FXR197 and RORg198. Furthermore, the microbiome may generate metabolites such as indole-3-propionic acid which are weak agonists for PXR199. However, the…"

Comment 8/ line 374: the problem of detecting compounds that bind but exert weak activation or antagonistic activity is not mentioned. Also other methods (biophysical for example) should be considered to overcome the limitations and problems encountered with eukaryotic cellular models.

Response: We missed this point. The idea of ligand-free based assays was to allow for the development of cell-based assays that detect weak agonists. The identification of weak agonists can be performed by addition of limited amounts of agonist to ligand-free systems. To this end we introduced the following phrase in lines 384-386:

“Ligand-free cell-based assays could be used to develop assays that detect weak agonists. Furthermore, ligand-free systems can be used to titrate the concentration of known agonist to develop screens for weak antagonists and inverse agonists.”

We understand that there are problems with cell-based assays and this is reflected by the phrase: “This may just reflect the fact that we do not know enough of the biology of some NHRs to define how to measure receptor activity.” In lines 380-381. Conversely, using biophysical methods is also problematic. These increase the rate of false-positives. This can be partially overcome by the affinity-tissue concentration rule. Another problem is that each class of ligand has completely different properties regarding their solubility in organic solvents and formation of micelles in water-based buffers. Thus, we added the following sentence to lines 409-419.

“The induction of reporters in cell-based assays can be indirect and the ligands identified in these assays need to be confirmed for direct binding to receptor. Early studies used very sensitive assays with radiolabeled ligand on cell extracts of mammalian cells transfected with glucocorticoid103, estrogen104 and retinoic acid receptors105,106. The production of recombinant protein from bacteria allowed one to measure NHR-ligand interactions using non-radioactive methods such as fluorescence polarization216, surface plasmon resonance217 and NMR218. Direct binding to NHR is a strong indication that a metabolite is a ligand. However, as direct screening methods they have their own limitations. Each ligand class has differences in solubility and chemical properties like formation of micelles in the buffers used for the assays. This requires ligand binding assays to be adjusted for the class of ligands being investigated and to be specific for fatty acyl amino acids 219, sphingolipids 170 or other lipid groups. Direct binding assays also increase the rate of false-positives and ligands detected using biophysical methods need to be confirmed by cell-based assays, binding affinity, tissue concentration and function “in vivo”.

We hope that these modifications are satisfactory to the reviewer.

Comment 9/ Typo: spaces to be deleted (lines 34, 192, 200, 322, 354).  

Response: we have eliminated these typos and any other typos we found during the revision. They have moved from the original line due to corrections in the text but we marked them in red.

Reviewer 2 Report (New Reviewer)

Comments and Suggestions for Authors

In this manuscript, Chen Y. et al. reviewed nuclear hormone receptors from evolutionary perspectives of receptors and their ligands. Manuscript is well written and interestingly summarized, but several points need to be corrected for publication.

1 In line 29, because first cloned NHRs were oestrogen receptor (Chambon P group) and glucocorticoid receptor (Evans RM group), “glucocorticoid and thyroid..” is better to change as “steroids and thyroid..”.

2 In line 32, I do not know the metabolite-dependent direct modification of CREB in nucleus, and ref. 4 does not mention on this topic. If reported, please cite other references. If not, delete “CREB”.

3 In line 51, ref.17 is not related to PPARgamma. Please change this ref. And as far as I was searching, there were no reports on the LBD deficient PPARG isoforms. They should add another reference, or delete it. Also, there are no reports on splicing isoform in NR4A3 after 1998 (ref19), so there is no need to mention it.

4 In line 111 of “Although, Houston et al 2022...”, they do not cite ref. Please cite this paper.

5 Regarding gene notation, NR notation and original gene names are mixed and difficult to read. For example, in line 115-116, “NR2A binds saturated...” is better to”HNF4s/NR2A” or NR2A/HNF4s. NR2C/TR2/4 may be better to NR2Cs(TR2/4). NR2E1 is better to NR2E1/TLX. Please check all notation and correct them.

6 In line 199, they should add estrogen receptor as first cloned NHR (Nature, 1986 Mar;320(6058):134-9.).

7 In line 245-246, pioglitazone is a PPARg selective ligand. Thus, PPARa should be deleted.

8 In line 249-251, what is the combination of NHR and another protein for type D system ? They should give a example of the type D system hypothesis. If not, delete this sentence or state that there is no example that demonstrates the type D system yet.

Author Response

Comment

In this manuscript, Chen Y. et al. reviewed nuclear hormone receptors from evolutionary perspectives of receptors and their ligands. Manuscript is well written and interestingly summarized, but several points need to be corrected for publication.

Comment 1 In line 29, because first cloned NHRs were oestrogen receptor (Chambon P group) and glucocorticoid receptor (Evans RM group), “glucocorticoid and thyroid..” is better to change as “steroids and thyroid..”.

Response: Thank you for the suggestion. We changed the text in line 29 to “steroid and thyroid…” as suggested by the reviewer.

Comment 2 In line 32, I do not know the metabolite-dependent direct modification of CREB in nucleus, and ref. 4 does not mention on this topic. If reported, please cite other references. If not, delete “CREB”.

Response: The reviewer is correct. CREB is indirectly regulated by metabolites through PKA. Other evidence of metabolites that directly bind to CREB Transcription factors is not strong. We therefore removed CREB from line 31-32 but left the PAS family. We replaced reference 4 for Lin et al Eur J. Med Chem, 2022 that contains examples of both natural and synthetic AHR (a PAS family member) ligands.

Comment 3 In line 51, ref.17 is not related to PPARgamma. Please change this ref. And as far as I was searching, there were no reports on the LBD deficient PPARG isoforms. They should add another reference, or delete it. Also, there are no reports on splicing isoform in NR4A3 after 1998 (ref19), so there is no need to mention it.

Response: We thank the reviewer for having noticed this error. A wrong citation was placed by mistake. We removed these citations as suggested and now the text in line 53 reads: “Furthermore, there are isoforms of the androgen receptor (NR3C4) lack a LBD domain18”.

As a request from reviewer #1 we extensively revised the legend for figure 1.

Comment 4 In line 111 of “Although, Houston et al 2022...”, they do not cite ref. Please cite this paper.

Response: we included the citation now on line 116.

Comment 5 Regarding gene notation, NR notation and original gene names are mixed and difficult to read. For example, in line 115-116, “NR2A binds saturated...” is better to”HNF4s/NR2A” or NR2A/HNF4s. NR2C/TR2/4 may be better to NR2Cs(TR2/4). NR2E1 is better to NR2E1/TLX. Please check all notation and correct them.

Response: We are sorry that our nomenclature has created confusion. We changed it and hope that it is much clearer now. Our changes are as follows: We are using the standard nomenclature when this is widely used. For receptors that are known by their common name we use the common name and in the first time we describe the receptor we add the standard name in parentheses. For example, the first time the thyroid hormone receptor appears in the text, it is designated THRA (NR1A1). Afterwards, the text contains only THRA since most readers are more familiar with THRA than NR1A1. Although NR2E1 is also known as TLX, both names are equally popular and we kept NR2E1. We added a table in figure 1 (panel F) which has the receptors with their standard and common names. In addition, to follow the official gene name, we decided to adjust the receptor names through the text: Thus, NHRs are a family, NR1 is a subfamily and NR1A is a group and NR1A1 is the receptor. The official gene name was provided in figure 1, panel F. The corrections for gene names are through the text and marked in red.

Comment 6 In line 199, they should add estrogen receptor as first cloned NHR (Nature, 1986 Mar;320(6058):134-9.).

Response: We thank the reviewer for having noticed this overlook from our part. The reference was added in now line 207-208.

Comment 7 In line 245-246, pioglitazone is a PPARg selective ligand. Thus, PPARa should be deleted.

Response: We removed PPARa from now line 256.

Comment 8 In line 249-251, what is the combination of NHR and another protein for type D system ? They should give a example of the type D system hypothesis. If not, delete this sentence or state that there is no example that demonstrates the type D system yet.

Response: we believe that adding the type D hypothetical system is important and clarified the text by adding: “There is no example of a type D system identified as of yet.” In now line 260-261.

Round 2

Reviewer 1 Report (New Reviewer)

Comments and Suggestions for Authors

I would like to thank the authors for taking my comments into account and for making the necessary corrections and additions. There are still disagreements between us on certain points, but these are more a matter of interpretation than fact. 

This manuscript is a resubmission of an earlier submission. The following is a list of the peer review reports and author responses from that submission.

Round 1

Reviewer 1 Report

Comments and Suggestions for Authors

This is a well-written review on nuclear hormone receptors (NHR), and a follow-up of the review provided by the same last author in 2020 (Cells. 2020 Dec 4;9(12):2606. doi: 10.3390/cells9122606.

Nuclear Hormone Receptors and Their Ligands: Metabolites in Control of Transcription). The manuscript contains very nice Figures.

 Comments:

In the abstract, a summary is given of the subjects addressed in this review. Below quotes from the abstracts are provided followed by comments:

1.       “….recent discoveries in the NHR field”;

The novelty of the content of the current review concerning NHRs is rather limited. The most recent papers cited (2021 to 2023) concern the identification of estrogen as a modulator of PPARj (ref #9-10), oleic acid as a ligand for NR2E1 (#38), sphingolipids and COUP-TF (#31), and 1-Oleoyl-lysophosphatidylethanolamine stimulating ROR activity (#44).

2.       “…propose new models of ligand-receptor co-evolution and emergence of hormonal function…”

This section is relatively long and has quite some overlap with the review mentioned above from the same group.

3.       “….models of regulation of NHR activity by canonical orthosteric ligands and the new emerging class of allosteric ligands…..”

This section is very short with limited novel insight.

4.       “…..models of autoregulation via feedback loops.”

The feedback loops for VDR and RAR have been known for a long time.

5.       “….we discuss the canonical assays that have been developed to identify NHR ligands…”

Luciferase/CAT-assays are rather old-fashioned and make this section of rather low interest.

6.       “….emerging new methodologies that could be used to identify natural ligands for human orphan NHRs….”

Here the authors describe the overexpression of enzymes that synthesize endogenous ligands may be an option to modulate the activity of NHRs, which is an interesting thought.

Other comments:

7.       p3, line 50:

Here the authors mention that “neither the DBD or LBD are essential”. This is a rather confusing statement. Indeed, NHRs without these domains have been identified, but for most NHRs the DBD is essential, given that these NHRs function as transcription factors that are fully dependent on their DNA-binding capacity. Similarly, NHRs with a known ligand are heavily dependent on binding their ligand with their LBD to become active.

8.       p.3, line 67

“Another example is glucose which is a physiological allosteric agonist for LXR”:

It may be better to leave out this example, since this observation has not been confirmed, and is doubted by the NHR community.

9.       Fig.3:

 What does the red star indicate? Please, adjust the legend.

10.   p.9, line 219:

In this paragraph, different ‘Types of systems A-D’ are presented. It is unclear where this classification comes from and is therefore rather confusing.

Reviewer 2 Report

Comments and Suggestions for Authors

The focus of the article is a bit vague. On the other hand, the general principle of NR activation is described, on the other hand ligands are mentioned, although not very comprehensive

In general, the text is largely difficult to read and frankly spoken, also tiring. Moreover, there is no clear focus of the article which makes it not very attractive for the reader. This is mainly due to the fact that the author cover a broad range of topics, which are at least partially covered in a very superficial manner. Topics include NR types, orthosteric and allosteric ligands, NR structure, receptor-ligand interactions, binding affinities, feedback loops, ligand identification and others.

The main part of the topic “How are NR ligands identified” does not have any clear relationship to NRs. The largest part describes the identification of hormones, but not how these were associated with NRs. In the same section the authors include a very simple description of reporter gene assays which can be used for investigating the effect of a compound on the NR activity.

Overall, the topic of the article is unclear and the content is largely written in a way that is too superficial to be useful for scientists.

For the reasons mentioned above, I cannot recommend the article to be published in this journal.

Minor comments:

Figure 2: the scheme is not necessary, it would be better to show X-ray crystal structures of the LBD in complex with orthosteric and allosteric ligands.

I know the article stating that 16 % of all approved drugs target NHRs, but I think this is not correct (any more) since only few drugs targeting NRs have been approved in the last 10 years (the cited article is from January 2017). Also see the Figure  in the article showing that only few NR-targeting drugs have been approved in the years before the article was released. Moreover, in the years since the article has been published only few NR-targeting drugs have been approved. This should be mentioned.

The overall text quality must be improved, there are many small errors and some sentences are unclear (see list below).

l. 114/115: Comment: C. elegans contains less than 1000 somatic cells and 1000-2000 germ cells.

Comments on the Quality of English Language

l. 6: are a family OF ligand-regulated

l. 8: NHRs2

l. 27/28: … and homology to glucocorticoid and thyroid hormone receptors, the founding members of the NHR superfamily (this part does not fit to the first part of the sentence)

l. 32: their ligands,

l. 43: (C) Class III

l. 44: bind TO direct repeats

l. 48: lack THE AF-1 domain

l. 51: PPARG is PPAR gamma or peroxisome proliferator-activated receptor gamma

l. 56: ligand-binding pocket (LBP),

l. 63: …these allosteric ligands are not physiological. (rephrase)

l. 66/67: is a physiological allosteric ligand (there is no physiologic ligand. I know what you mean but it is not written correctly). Similar in the following sentence.

l. 78: ligand-binding domain

l. 81: contains

l. 81/82: …protein, AqNR1 and AqNR2.

l. 177: ligand-modulated.